# Effects of the Response to the COVID-19 Pandemic on Assault-Related Head Injury in Melbourne: A Retrospective Study

**DOI:** 10.3390/ijerph20010063

**Published:** 2022-12-21

**Authors:** Juan F Domínguez D, Johnny Truong, Jake Burnett, Lata Satyen, Hamed Akhlaghi, Julian Stella, Nick Rushworth, Karen Caeyenberghs

**Affiliations:** 1Cognitive Neuroscience Unit, School of Psychology, Deakin University, Geelong, VIC 3220, Australia; 2School of Psychology, Deakin University, Burwood, VIC 3125, Australia; 3Department of Emergency Medicine, St. Vincent’s Hospital, Melbourne, VIC 3065, Australia; 4Department of Emergency Medicine, Geelong Hospital, Geelong, VIC 3220, Australia; 5Brain Injury Australia, Sydney, NSW 2112, Australia

**Keywords:** head injury, COVID pandemic, traumatic brain injury, domestic violence, random assault, physical assault

## Abstract

Assault is the leading preventable cause of death, traumatic brain injury (TBI), and associated mental health problems. The COVID-19 pandemic has had a profound impact on patterns of interpersonal violence across the world. In this retrospective cross-sectional study, we analysed medical records of 1232 assault victims (domestic violence: 111, random assault: 900, prison assault: 221) with head injuries who presented to the emergency department (ED) at St Vincent’s Hospital in Melbourne, Australia, a city with one of the longest and most severe COVID-19 restrictions worldwide. We examined changes in prevalence in the assault group overall and in domestic violence, random assault, and prison assault victims, comparing data from 19.5 months before and after the first day of COVID-19 restrictions in Melbourne. Moreover, we investigated differences driven by demographic factors (**Who**: age group, sex, and nationality) and clinical variables (**Where**: assault location, and **When**: time of arrival to the ED and time from moment of injury until presentation at ED). Descriptive statistics and chi-square analyses were performed. We found the COVID-19 pandemic significantly affected the **Where** of assault-related TBI, with a shift in the location of assaults from the street to the home, and the increase at home being driven by random assaults on middle-aged adults. Overall, we observed that 86% of the random assault cases were males, whereas 74% of the domestic assault cases were females. Meanwhile, nearly half (44%) of the random assault victims reported alcohol consumption versus a fifth (20%) of domestic violence victims. These findings will have direct implications for developing screening tools and better preventive and ameliorative interventions to manage the sequelae of assault TBI, particularly in the context of future large-scale health crises or emergencies.

## 1. Introduction

Physical assault, by which we mean any intentional and unlawful physical force applied by a person to the body of another person [1], is a leading preventable cause of death, traumatic brain injury (TBI), and associated mental health problems, particularly amongst women and vulnerable populations. The COVID-19 pandemic and associated restrictions have had a profound impact on patterns of interpersonal violence across the world [2], be it domestic violence (DV), which in this paper includes family violence (FV), gender-based violence, violence against women and intimate partner violence (IPV), or violence from strangers or nonfamily acquaintances (which we refer here as random violence, (RV)). Therefore, there is an urgent need to understand the effect of the pandemic on assault-related TBI.

Two and a half years from the start of the pandemic, the picture that emerges of the effect of COVID-19 on DV is a complex one, particularly when focusing on physical assault. Regarding DV, owing to the potential confinement and increasing isolation of victims with violent partners brought about by the restrictive measures adopted to manage the pandemic, the United Nations (UN), barely two weeks after the World Health Organization declared COVID-19 a pandemic, issued a statement warning that this form of violence could escalate. The statement urged governments around the world to renew their efforts to minimise the increasing risks of violence [3]. In April 2020, Phumzile Mlambo-Ngcuke, Executive Director of UN Women, referred to violence against women as the ‘shadow pandemic’ [4], citing reports of increasing DV from helplines and shelters from various countries. Consistent with this, systematic reviews have reported an increase in DV in several jurisdictions across the world since the start of the pandemic [5,6,7,8].

There is still, however, substantial variability in the findings reported by these reviews. For example, Piquero and colleagues (2021) found an increase in DV in 29 out of 37 instances reported, with the remaining eight exhibiting a decrease [6]. Both Bazyar et al. (2021) and Lausi et al. (2021) reported an increase in IPV [5,7]. In the case of Lausi et al. (2021), however, the increase was observed when including both physical and nonphysical instances of violence (e.g., sexual, economic, psychological) [7]. Considering physical IPV on its own, while the severity was reported by victims to increase, the number of physical assaults decreased. Moreover, results from a more recent review were inconclusive regarding the prevalence of physical DV during COVID-19, with studies reporting both increases and decreases. This review paper also showed a change in severity of psychological/emotional and sexual DV during the COVID-19 pandemic [9].

A closer look at the literature suggests that when considering all reported cases of DV, there were only slightly more instances where DV increased versus decreased following the introduction of COVID-19 restrictions. Moreover, when focusing on cases of DV more likely involving physical assault (i.e., from worldwide studies using police crime data, rather than calls to hotlines), the number of reported instances where DV decreased or stayed the same outnumbered the instances where DV increased [10,11,12,13,14,15,16,17,18,19,20,21,22,23,24,25,26,27,28,29,30,31]. This finding, however, could be due to victim-survivors’ reduced ability to report an incident of DV and access services during the pandemic [32]. This is consistent with previous findings showing that law enforcement is frequently the last resort for DV victims who often choose community-based services such as emergency hotlines before reaching out to law enforcement [33]. Moreover, the mixed findings in prevalence estimates for DV can also be attributed to the differences in data collection.

Regarding RV, COVID-19 restrictions led to no change or a reduction in reported incidents [10,11,13,15,16,19,24,25,29]. In studies that did not distinguish between DV and RV, most cases exhibited a decrease in assaults, followed by no change [14,28]. This pattern was the same including or excluding hotline or survey data.

The above findings show substantial heterogeneity in the effect the COVID-19 pandemic had on interpersonal violence, particularly DV. In addition to demographic and socioeconomic factors, there is considerable variability across jurisdictions in the timing, severity, and geographic coverage of policies that countries have implemented to manage the pandemic, including policies pertaining to containment and closure strategies, economic responses, health system programs, and vaccine roll-out [34,35]. Government responses to COVID-19 ranged, for example, from keeping society mostly open, as in the case of Sweden, to a ‘zero community transmission’ approach adopted in countries such as Australia, New Zealand or Vietnam, prior to achieving vaccination targets.

Australia is a particularly interesting case, as it offers the opportunity to evaluate the effect of one the longest and most stringent regimes of COVID-19 restrictions in the world (see Table 1), on interpersonal violence and more specifically, assault. Figure 1, for example, shows how mobility was largely restricted to the home, grocery stores, and pharmacies, and substantially reduced to workplaces, retail, and recreation venues, parks, and transit stations starting in March 2020, and mandatory COVID-19 isolation periods ended in Australia on 14 October.

Initial evidence suggests that the pandemic restrictions in Australia have had an effect on interpersonal violence. Despite 2020 being referred to by the media as ’the worst year for domestic violence’, and support services reportedly struggling to meet demand [36], the Australian case is, similarly to the rest of the world, more complex. In the state of Victoria, studies have variously reported an increase in violence against women (via survey of help professionals), family common assaults and FV incidents (police crime data), and assault-related injuries at home (emergency department (ED) records) in both male and female victims [13,31,37]. However, a decrease in Victoria was reported for serious domestic assaults related to FV [13] and in family-related assaults (overall, without distinguishing severity) [25]. Moslehi and colleagues found that the decrease in family-related assaults was almost half of the decrease in other assaults [25]. Moreover, their analysis revealed that this was only true for local government areas in Victoria below the median income, further illustrating the role of place-related variability in the effect of pandemic restrictions [25]. However, again, the decrease may not present a true picture of the extent of domestic violence experienced, due to under-reporting or differences in data sources.

National statistics show that situational stressors during the pandemic possibly exacerbated the main drivers of violence and increased the rates, complexity, and severity of violence [38], while other reports show, in contrast, a decrease in domestic assaults in New South Wales (police crime data), and no change in breaches of DV orders (official offence rates) in Queensland [13,25,29]. Many women also experienced violence for the first time during COVID-19 [38]. Those who had previously experienced violence reported increased rates of physical and sexual abuse and emotionally abusive, harassing, or controlling behaviours during COVID-19; several women also reported that they were unable to seek assistance from services during this period [38]. Regarding RV, all available Australian studies reported a decrease in assault overall, but also in common and serious assault [13,25,29].

As noted at the outset, there is an urgent need to investigate the effects of COVID-19 restrictions on physical assault, particularly when it is most serious and may result in traumatic brain injury. While TBI ranges in severity (80% of TBI patients are mild [39]), it can result in various long-term physical, cognitive and emotional sequelae [40,41,42]. Moreover, the pathophysiology of assault TBI is likely unique compared to other TBI causes due to the coupling of the physical and psychological trauma of intentional violence. However, studies investigating the impact of the COVID-19 crisis on assault-related TBI are sorely missing. Data from presentations to EDs can provide unique insight into this problem as they comprise information regarding location and type of injury.

Another gap in the literature is the effect of COVID-19 restrictions on physical assaults in places with the most severe and longest restrictions. Most of the studies focus on the months of March to June 2020. The city of Melbourne, Australia, offers an ideal case to assess the sustained effect of the COVID-19 pandemic restrictions on assault-related TBI: Melbourne was the place with the second longest lockdown in the world by the end of 2021 (263 days across six separate lockdowns [43]), in addition to other restrictions and stay at home orders spanning from 16 March 2020 to 21 October 2021.

This study investigated changes to the **Who, Where,** and **When** of assault-related TBI brought about by COVID-19 in Melbourne using ED presentation data from St Vincent’s Hospital, an inner-city hospital in Melbourne. Regarding **Who**, the paper studied changes in prevalence and demographics in the assault-related TBI population, comparing data from 19.5 months before and 19.5 months after 16 March 2020 (the first day of restrictions in Melbourne). We examined changes in the assault TBI group overall, but also in subgroups of assault victims (DV victims and RV assault victims), and vulnerable populations (prison population and Aboriginal and Torres Strait Islanders). We also investigated differences driven by demographic factors including age group, sex, and nationality. The study also tracks changes in the assault location, **Where**, and the timing of victims presenting to ED, **When**. Finally, we report on the data without partitioning into pre-COVID and COVID periods, to provide an overall picture of interpersonal violence across the whole period.

## 2. Methods

### 2.1. Study Design

This study utilised a retrospective cross-sectional design of all patients with assault-related head injury (as an indicator of likely TBI) who were admitted to the Emergency Department (ED) of St Vincent’s Hospital in Melbourne (SVHM). SVHM is a comprehensive adult health service for people aged 18 years and older who live in the inner urban east area of Melbourne. SVHM catchment includes vulnerable populations (such as Aboriginal and Torres Strait Islanders and prisoners), which are generally more disadvantaged and have higher health care needs than the wider Australian population. There is currently limited data available to understand the extent to which COVID-19 has impacted prisoners’ health in particular. We therefore decided to include this vulnerable group in the present study. Of note, there were no changes in policy by the ED regarding accepting patients relative to their injury severity given the challenges associated with the pandemic and the health care system’s response to the pandemic. The patients were identified from the electronic medical record system of SVHM (H.A.). Their data were extracted from this medical record system and nonidentifiable data collected in an independent database for further analyses. Ethics approval for the study was obtained from St Vincent’s Hospital Human Research Ethics Committee (Project ID: 66135, LRR 167/20).

### 2.2. Study Population

Figure 2 depicts an overview of the inclusion/exclusion criteria to select patients for our study sample. Any patient admitted from the ED who met the following criteria was included in the present study: (i) aged 18 or older; (ii) ED discharge date between 30 July of 2018 and 30 October of 2021; and (iii) presenting with ‘head injury’ using the injury descriptor, or ED diagnosis using International Classification of Disease Tenth Revision (ICD-10) codes (namely S00–S09.2 and S09.7–S09.9). Patients were excluded if: (i) the head injury occurred outside Victoria (e.g., transferred to SVHM from other states); or (ii) the head injury was a result of police intervention or self-harm. Some patients had multiple ED presentations resulting from the same assault event. In this case, we only retained the first presentation and discarded subsequent presentations. In the case of subsequent admissions for new injuries of different assault events, we included these cases in our study population.

In the present study, we categorised the assault patients with head injury into three subgroups: (i) RV assault (defined as assault by a stranger or an acquaintance that is not a family member or prison mate), (ii) DV assault (involving the assault by a partner, ex-partner, or family member), and (iii) prison population (PP) assault. We utilized three descriptors in the database to distinguish the three assault subgroups and avoid ambiguity. Specifically, we used (i) the human intent descriptor, which described whether the aggressor was an intimate partner; (ii) the description of the injury event, which provided a short description of the assault; and (iii) the presenting complaint comments, which encompassed accompanying narratives from patients written by the triage nurses. Importantly, several assault cases were incorrectly categorised as unintentional head injury (see Figure 2). These cases were included in the analysis if the corresponding ED presenting complaint comments provided unambiguous mention of assault (~1.5% of cases were therefore excluded for this reason). ED presenting complaint comments were thoroughly checked in every single case in the dataset. Data cleaning and quality control (including correction of miscategorised cases) was performed by J.T. under the close supervision of clinician H.A. and in consultation with K.C. and J.F.D.D.

As can be seen in Figure 2, 5873 patients were presented with a head injury (using the ICD-10 diagnosis codes) at the ED of SVHM during the 39-month data collection period. Of these head injury patients, 1232 were due to assault, with 900 RV assault cases, 111 DV assault cases (9%), and 221 cases (18%) occurring in PP. Within RV and DV assault subgroups, approximately 20% of included cases had incorrect human intent descriptors and needed to be corrected and recategorised.

To investigate changes to the **Who**, **Where,** and **When** of assault-related TBI brought about by COVID-19, we subsequently extracted demographic variables from the selected sample. Concerning the ‘Who’, we obtained the following demographic data: (i) sex; (ii) age group, which here we define as younger adults (18–39), middle-aged adults (40–59), and older adults (60+) (in close agreement with the literature [44,45,46]); (iii) nationality, categorised as Australian, Aboriginal, and Torres Strait Islander Peoples, non-Australian, or unknown; and (iv) alcohol consumption (self-reported). Regarding ’When‘ and ’Where‘, we collected time of arrival to the ED (grouped into four time windows, i.e., morning (6:00 a.m.–12:00 p.m.), afternoon (12:00 p.m.–6:00 p.m.), evening (6:00 p.m.–12:00 a.m.) and night (12:00 a.m.–6:00 a.m.), whether patients were admitted to the ED within 24 h of injury and the location where the head injury was sustained (home, road/street, place for recreation, workplace and other public areas, prison, unspecified).

### 2.3. Data Analysis

Descriptive statistics were calculated for all the demographic and injury-related variables. Specifically, frequency tables were computed for age group, sex, nationality, alcohol consumption, and time of injury. These frequencies were stratified by assault subgroups (i.e., RV, DV, and PP assault victims). Chi-squared analyses were conducted to examine changes in the frequency of assault cases before and during the COVID-19 pandemic. Differences in variables between these time periods and across assault subgroups were also explored with chi-square tests. Specifically, we categorised the data from 19.5 months before and 19.5 months after 16 March 2020. An additional chi-squared analysis was performed explicitly and specifically investigating if lockdowns (as different from all COVID-19 restrictions), differentially affected the proportion of assault across assault groups. Cases during the lockdowns were compared to cases during the equivalent periods before the pandemic, in 2019. We focused our analysis on lockdowns 1 and 6 as these represent lockdown periods at the beginning and end of COVID-19 restrictions and as they lasted for a substantial amount of time. Specifically, lockdowns 1 and 6 lasted 43 and 77 days respectively, as opposed to lockdowns 3, 4, and 5, which lasted between 5 and 14 days. We drew cases in 2019 from 30 March to 12 May (as the equivalent period for lockdown 1), and 5 August to 21 October (as the equivalent for lockdown 6). Statistical analyses were conducted using JASP Version 0.16.3 (JASP Team, 2022).

## 3. Results

### 3.1. Effect of the COVID-19 Pandemic

#### 3.1.1. Who Were Victims of Assaults with Head Injuries in Melbourne during the COVID Pandemic?

A total of 689 cases of assault resulting in head injury were recorded in the pre-COVID period, compared to 543 cases during the COVID pandemic (see Figure 3 for an overview). This represents a 21.2% decrease in these cases. Overall, the COVID pandemic did not significantly affect proportions of cases by assault subgroup, sex, age, or nationality (for all variables, *p* > 0.05). When restricting analysis to lockdown periods 1 and 6, we also did not find any differences in proportions of cases by assault subgroup compared to the corresponding period in 2019, prior to the pandemic (*p* > 0.10). However, we observed a significantly lower number of middle-aged PP assault victims (aged between 40–59 years) during the COVID period compared to the pre-COVID period [*χ^2^* (1) = 9.10, *p* < 0.01; Figure 4] (there were not enough individuals aged 60 or older for this test). Frequencies of demographic variables by COVID period and stratified by assault subgroup are presented in Table 2.

#### 3.1.2. Where Were Victims of Assaults with Head Injuries during the COVID Pandemic?

We found that the location of injury significantly differed between the pre-COVID and COVID periods [*χ^2^* (3) = 23.78, *p* < 0.001, Cramer’s V = 0.17]. Examination of the adjusted residuals (FWE corrected) revealed that home (3.82, *p* < 0.001) and road/street/highway (−3.14, *p* < 0.01) contributed significantly to this result, with home exhibiting an increase in assault-related TBI victims during the COVID period, compared to the pre-COVID period, and the inverse occurring for assault cases occurring on the streets (Figure 5A). No pandemic effect was found between assault subgroups (random and domestic victims) at home or on the streets [χ^2^ (1) = 0.22, *p* > 0.05]. This 2 × 2 × 2 cross tabulation analysis had one cell with an expected frequency less than 5. For these two locations, post hoc analyses of sex and age across assault subgroup and time period revealed only an effect on middle-aged adults at home [χ*^2^* (1) = 5.58, p < 0.05, Cramer’s V = 0.31], with increased random assault cases and decreased domestic violence victims observed during COVID-19 (Figure 5B). Frequencies of the locations of assault injuries by COVID period and stratified by assault subgroup are presented in Table 3.

#### 3.1.3. When Did Victims of Assault-Related TBI Present at the ED during the COVID Pandemic?

The time of day in which assault patients arrive to the ED was unaffected by the COVID pandemic across all assault subgroups (*p* > 0.05). There was also no pandemic effect on whether assault patients arrived at the ED within 24 h (*p* > 0.05). None of the assault subgroups revealed any significant differences in COVID period for these time variables either (*p* > 0.05). Frequencies of time variables by COVID period and stratified by assault subgroup are displayed in Table 4.

### 3.2. Overall Results Independent of COVID-19

As the pandemic effect on assault-TBI was limited to a prison age subgroup and location, we report results after collapsing across both pre-COVID and COVID periods, from 30 July 2018 to 30 October 2021.

#### 3.2.1. Who Were Victims of Assaults with Head Injuries?

Of the 1232 patients with a head injury resulting from an assault, 900 (73.1%) were due to RV assault, 111 (9%) due to DV assault, and 221 (17.9%) were PP victims (see Figure 6 and Appendix A). Assault victims were predominantly male (n = 1016, 82%), aged between 18–39 years old (n = 842, 68%), and of Australian nationality (n = 743, 60% excluding unknown cases; see below and Appendix A). Six percent (n = 76) of Australian assault victims were of Indigenous background. A third of the victims reported having consumed alcohol at the time of injury (n = 325, 33% excluding unknown cases).

There was a clear contrast in the sex of the victim between RV and DV assaults: 86% of the RV assault cases were males, whereas 74% of the DV assault cases were females. Most victims across all groups were young adults (68% overall). In the general population, approximately 60% of victims were Australian and 6% were Aboriginal or Torres Strait Islanders. In the prison population, 74% of the victims were Australian and 7% Aboriginal or Torres Strait Islanders. Another interesting difference between RV and DV assaults related to alcohol consumption, with nearly half (44%) of the RV assault victims reporting alcohol consumption versus a fifth (20%) of DV assault victims.

A total of 19.1% (235) of all cases did not have information regarding alcohol consumption and 5% of all cases (61) did not register their nationality.

#### 3.2.2. Where Were Victims of Assaults with Head Injuries?

Excluding PP victims, the majority of victims were assaulted in the street (38%), followed by the workplace or other public areas (29%), home (22%), and places of recreation (11%) (Figure 7 and Appendix A). RV assaults followed a similar pattern: the street (41%), followed by the workplace/public areas (33%), then by home and places of recreation (both at 13%). In contrast, DV assaults took place, as expected, overwhelmingly at home (87%), followed by the street (9%) and the workplace/public places (4%), and none taking place in a place of recreation. In addition, the location of 18% (n = 222) of all assaults was unknown (see Appendix A).

#### 3.2.3. When Did Victims of Assault-Related TBI Present at the ED?

The number of victims of assault increased across the morning (6:00–12:00), afternoon (12:00–18:00), and evening (18:00–24:00) periods in all assault subgroups, and it further increased at night (0:00–6:00) for RV assaults, when it peaked (Figure 8 and Appendix A). In contrast, the number of DV and PP assaults decreased during the night period.

In addition, 85.6% (n = 617) of assault patients reported arriving at the ED within 24 h of the assault and 14.4% (n = 104) reported arriving 24 h after the assault. Time of injury information was unknown for 511 individuals (41% of all cases; see Appendix A).

## 4. Discussion

### 4.1. Effect of the COVID-19 Pandemic

In this paper, we investigated changes in the **Who**, **When,** and **Where** of assault-related TBI brought about by the COVID-19 restrictions in an inner-city hospital in Melbourne, Australia. Overall, we observed a 21% decrease in the number of assault-related TBI. This is in line with Australian [13,25,29] and international trends showing a reduction (and no change is some instances) in rates of assault, overall and regarding random assault [10,11,13,14,15,16,19,24,25,28,29]. Our finding of a decrease in domestic assault, on the other hand, adds to the heterogeneity in this category, which is divided between reports of increases and decreases in cases (as can be seen in Appendix A), with slightly more increases [10,11,12,13,14,15,16,17,18,19,20,21,22,23,24,25,26,27,28,29,30,31,38]. For example, Moslehi and colleagues (2021) utilized police data and revealed a decline in domestic assault victims during the lockdown in New South Wales and Victoria [25]. In contrast, an analysis of Victorian health practitioners’ responses to a survey revealed that the COVID pandemic has led to an increase in the frequency and severity of violence against women [37]. These mixed findings in the literature may reflect the different methods used to collect the data (e.g., ED admissions, reports of calls to DV shelters, police calls, surveys of health practitioners, crime data). This decrease in assault-related TBI cases is also in accordance with ample studies showing significant declines in ED attendance during the pandemic (in keeping with stay-at-home orders), but also influenced by fear of acquiring COVID-19 or avoiding hospitals [47,48]. The true number during the COVID pandemic is therefore likely underestimated as assault victims might not have sought help [38] or received medical care (including hospital attendance) for their head injury. This can result in assault survivors living with an undiagnosed head injury, which can have devastating long-term health consequences if untreated [49]. This issue has been raised by other studies examining ED presentations during COVID-19 [31]. Besides misclassifications or under-reporting of DV, the random assault-related TBI cases at home could be potential intrafamilial violence. Indeed, in Victoria, 1138 incidents of adolescent family violence were recorded in 2018–19 [50,51], with young aggressor violence increasing by 11.8% over the past five years [50]. We also know that there is under-reporting of this violence due to victims who are family members (mostly women) and feeling protective of the young person and not implicating them in the assault or seeking further help for it [52].

We found no effect of the COVID-19 restrictions on the **Who** and **When** of assault-related TBI. There was therefore no difference in the effect of COVID-19 restrictions between RV, DV, and PP assault groups, or between males and females, or according to age group, nationality, or alcohol consumption. However, we found the COVID-19 pandemic significantly affected the **Where** of assault-related TBI. Specifically, we observed a shift in the location of assaults from the street to the home. The significant increase in intentional head injuries sustained at home during the COVID-pandemic can be explained by theories of aggression [53] in psychology, or strain [54] in criminology. According to the general aggression model (GAM) [53], for example, there are multiple causal proximal factors (including high levels of frustration and stress) and distal factors (such as difficult life conditions) that may result in an increase in assaults in the home environment.

Pandemics requiring quarantining have been shown to bring about a wide range of stressors (e.g., social isolation, diminished physical and daily activity and routines, fears of infection, financial insecurity, job loss, inadequate supplies and information, reduced access to health services) resulting in negative psychological effects (e.g., depression, stress, post-traumatic stress disorder, anger, sleep disorders, and problematic substance use) [55], present also during the COVID-19 pandemic [56]. Together, these stressors and associated psychological effects may amplify the risk of violence [57,58,59]. In Australia, unemployment rose from 5.2% in March to 6.8% in August 2020 [60]. Additionally, the levels of stress and depression were shown to increase by 25% among Australians in the same year after the onset of the pandemic [61]. The shift of assault to the home during COVID-19 is consistent with previous work revealing that more assault injuries requiring hospital treatment occurred at home during 2020 compared to 2019 in both regional and metropolitan Victoria [31]. Additionally, in line with VISU’s findings (VISU, 2020; editions 1–9) of an increase in intentional injuries at home irrespective of sex, we found no difference in the effect of COVID-19 restrictions on assault-related TBI between males and females. Our observed decreases in assault-related TBI cases sustained on the street are in line with findings suggesting fewer injuries were sustained in public spaces during the COVID pandemic [62]. This is also in accordance with opportunity theory and routine activity theory, which predict a fall in levels of crime occurring in public spaces (including assault) due to disruption in daily patterns of mobility of potential victims, guardians, and perpetrators [24,28].

Interestingly, the increase in assault-related TBI cases at home was driven by RV assaults (i.e., physical attack by a stranger or an acquaintance that is not a family member) on middle-aged adults (40–59 years old). While these assaults were classified as random (as opposed to domestic violence or prison assault), this categorisation was based on the testimony provided by patients as recorded by triage nurses in the human intent descriptor and ED presenting complaint comments. Unless explicitly stated by patients, no other information was available indicating the perpetrator of these assaults. Thus, it is possible that these particular assault-related TBI cases (perpetrated by an unspecified assailant) could be domestic assaults otherwise misclassified, due to the lack of accuracy of the self-report by the victims [49]. It is not readily apparent why the shift in the location of assault was driven by middle-age adults, and interpretation is further limited as there were not enough older adults to include in the analysis.

Self-reported history among domestic violence victims may be further confounded by both under- and/or over-reporting of brain injury incidence, due to the potential for misunderstanding specific terminology, emotional avoidance, and poor recall or suppression of event recall. Suppression may flow from the potentially traumatic nature of the experience (e.g., the development of post-traumatic stress disorder (PTSD) symptomatology that may interfere with recall), or lack of awareness of or minimisation of the severity of the injury (e.g., due to fear of potential repercussions of reporting or distrust of authority figures) [49].

Finally, this is the only study reporting on the impact of the pandemic and associated restrictions on the prison population. Our subgroup analysis revealed a significantly lower number of middle-aged prison assault victims during the COVID period compared to the pre-COVID period.

### 4.2. Overall Results Independent of COVID-19

Looking at the data without partitioning into pre-COVID and COVID periods, we observed noticeably contrasting differences in the proportion of cases between random assault and domestic violence in terms of sex and alcohol consumption. Specifically, 86% of the random assault cases were males, whereas 74% of the domestic assault cases were females. Nearly half (44%) of the random assault victims reported alcohol consumption versus a fifth (20%) of domestic violence victims. These patterns are in line with the general aggression model, which posits alcohol use as a situational risk factor and gender as a biological distal factor that might contribute to aggressive behaviour [53].

Within our domestic assault population, a significantly greater proportion was women (74%). This aligns with prior studies examining brain injury in domestic violence victims [63,64] and reviews of a range of injuries resulting from physical assault, including head, neck, and facial injuries [65]. Although both men and women experience domestic violence [66], women are (i) at greater risk of family, domestic, and sexual violence; (ii) more frequently injured and experience more severe injuries, in comparison to male partners [67]; and (iii) less likely to seek help for their physical and mental health needs [68,69]. The high prevalence rates of violence against women in this study align with other Australian and international studies. For example, globally, one in four ever-partnered women has experienced physical or sexual violence from an intimate partner since the age of 15 years [70]. In Australia, one in six Australian women (compared to 1 in 16 men) has experienced physical or sexual violence from a male relative since the age of 16 years [71]. Domestic violence victims who suffer from head injuries often experience sequelae, such as decreased cognitive functioning, memory loss, and post-traumatic stress disorder [49]. This indicates the urgent need of preventive measures for family and domestic violence that may include a combination of government and nongovernment policy initiatives, and changes in public discourse and practice.

Women are most likely to know the perpetrator (often their current or a previous partner) and the violence usually takes place in their home. In contrast, men are more likely to experience violence from strangers and in a public place [71]. Our results showed that Australian males injured by random assault presented as the group of patients at highest risk of assault TBI across the time periods, which is supported by previous retrospective studies [72,73,74,75]. To our knowledge, very little research has been conducted on the extent and impact of head injuries incurred during random assaults [76]. This highlights the need for targeted treatment approaches for this subgroup. Clinicians should be provided with established treatment guidelines and educational resources to improve their awareness of the many types of violence, how to identify and assist victims, and where to refer these patients for follow-up treatment in their community. Particular attention will need to be paid to addressing the treatment barriers faced by male victims that present with assault-related injury. For example, research on patterns of health service use suggests that men utilise healthcare services and engage in treatment follow-up less frequently than women [77,78].

Various internal factors (e.g., embarrassment, masculinity norms, cultural norms, negative attitudes towards help-seeking, etc.) and systemic barriers (e.g., accessibility issues, lack of time, ED waiting times, etc.) have all been reported as obstacles that discourage men from seeking medical help [79,80,81]. Moreover, innovative public health promotion strategies will need to be developed that aim to improve health literacy and treatment–engagement behaviours in men, and emphasise preventative health measures (i.e., reducing male violence), particularly in male-dominant settings (e.g., pubs, sports bars).

Furthermore, these violence prevention strategies should focus on reduction in alcohol use. Our results showed that alcohol consumption appeared to play an important role in assault-related head injuries in Melbourne. Alcohol was present in 44% of the random assault victims, and 20% of the domestic violence victims. Other retrospective studies have found a similar relationship between alcohol consumption and violence (e.g., Romania [72], Norway [82]). A recent review [83] suggests that brief alcohol intervention programs delivered in emergency department settings can be a cost-reducing approach to treating excessive alcohol consumption and improve heath in the long-term. Another strategy would be to limit drinking hours (and restricting availability of alcohol), which has showed to be effective in some countries [84].

In broad agreement with Australian assault data [85,86], we found for head injury victims in our study that: (i) most were young adults, followed by middle-aged adults, and older adults; (ii) most were born in Australia, rather than overseas; and (iii) were most commonly assaulted in a nonresidential location. The proportion of Aboriginal and Torres Strait Islander peoples in our sample with a head injury in the 2018–2021 period (~10%) was substantially less than the proportion of Aboriginal and Torres Strait Islander peoples hospitalised due to assault in 2019–2020 in the whole of Australia (~31%) [87]. This latter figure can only be considered indicative, as it is a national figure that refers to hospitalisations broadly, rather than solely due to a head injury. One final result that may be of interest for ED practitioners is the time of admission for intentional head injuries, which we found to increase across the day and peak in the evening period for DV and PP assaults. For RV assaults the time of admission similarly increased throughout the day but, in contrast with DV and PP, it peaked in the night period.

### 4.3. Limitations

While this study provides one of the first reports to examine the effect of the COVID-pandemic on assault-related head injuries in Australia, it has some limitations. Firstly, it is important to keep in mind that in this study we focus on assault-related head injuries. Our findings are therefore indicative of a particularly serious form of violence rather than more broadly being reflective of less severe types of physical violence or alternative forms of violence (like emotional, verbal, or financial abuse). Second, our design was cross-sectional and retrospective. The data retrieved can only suggest but not demonstrate a causal relationship [88]. Future studies examining the impact of a pandemic on intentional head injuries should adopt a prospective, longitudinal design, to allow the identification of risk factors and examination of causal links between variables [89]. These longitudinal studies would also offer great utility for the validation of theories of aggression [53] in psychology, or strain [54] in criminology. Moreover, this study was conducted in a single adult hospital of the inner eastern region of metropolitan Melbourne (St Vincent’s Hospital) and this could limit the generalizability of the findings to young populations (including children and adolescents) or to other parts of Australia (including rural and regional areas). Future multicentre studies should endeavour to gather medical record data from a range of emergency departments across age groups and different sites, as was done by Cassell et al. [90]. On the other hand, as detailed in the introduction, there is much variability in the effect of COVID-19 restrictions on interpersonal violence, so studies focusing on specific places and age groups are crucial to understanding this variability.

Another limitation is related to missing data in the medical record database, including detailed clinical characteristics of the head injury (e.g., severity of traumatic brain injury), and this limited the data analysis. In addition, several variables (alcohol consumption, nationality, location of assault) were left unspecified in some cases, which could alter the results. Also introducing a level of uncertainty in our findings is a proportion of assault cases (20%) that were miscategorised and needed to be corrected. Furthermore, patients with multisystem trauma, in addition to trauma to the head, might not have been categorised as having a head injury and could therefore be missing (e.g., chest stabbings with additional head trauma are more likely to not be categorised as TBI). Clinician entries for our sample exist (that were beyond the purview of this study), which could reveal further inconsistencies and help further confirm existing cases. Future work could establish a statewide or national injury registry system and mandatory recording of all clinical characteristics, to manage gaps and sources of uncertainty in the recording of important clinical information. This will be crucial to better understanding the extent of the problem of head injuries in assault victims.

## 5. Conclusions

In this study, we found the COVID-19 pandemic significantly affected the **Where** of assault-related TBI. We saw assaults causing head injury shifted from the street to the home, with the increase at home being driven by random assaults on middle-aged adults. Our findings provide an understanding of the effects of sustained COVID-19 restrictions on interpersonal violence, with novel insights into assault-related head injuries. It is expected that these findings will guide the development of novel screening tools, targeted preventive measures, and intervention strategies, and by doing so it will inform policy and practice to enhance the mental health outcomes of assault victims living with TBI.

## Figures and Tables

**Figure 1 ijerph-20-00063-f001:**
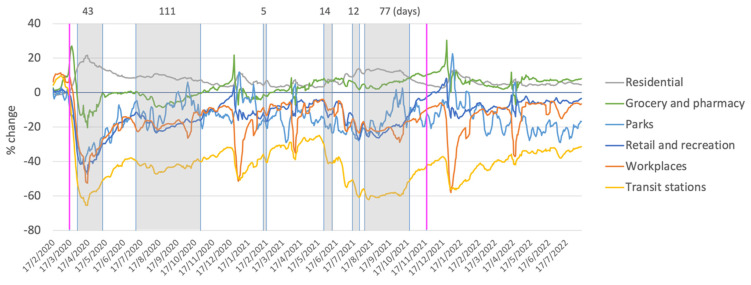
Changes in community movement in specific locations in Australia relative to the period before the COVID-19 pandemic. The ‘Residential’ category shows percent change in duration of time spent at home; the other categories measure a change in total visitors (compared to baseline days: the median value for the 5-week period from 3 January to 6 February 2020). First (16 March 2020) and last (19 November 2021) days of restrictions are indicated by magenta reference lines. Shaded areas represent Melbourne lockdowns. The duration of each lockdown (in days) is indicated above each lockdown period (Note: Mandatory COVID-19 isolation periods ended in Australia on 14 October 2022). Source: Google COVID-19 Community Mobility Trends–last updated 16 August 2022 (https://www.google.com/covid19/mobility/ accessed on 18 August 2022).

**Figure 2 ijerph-20-00063-f002:**
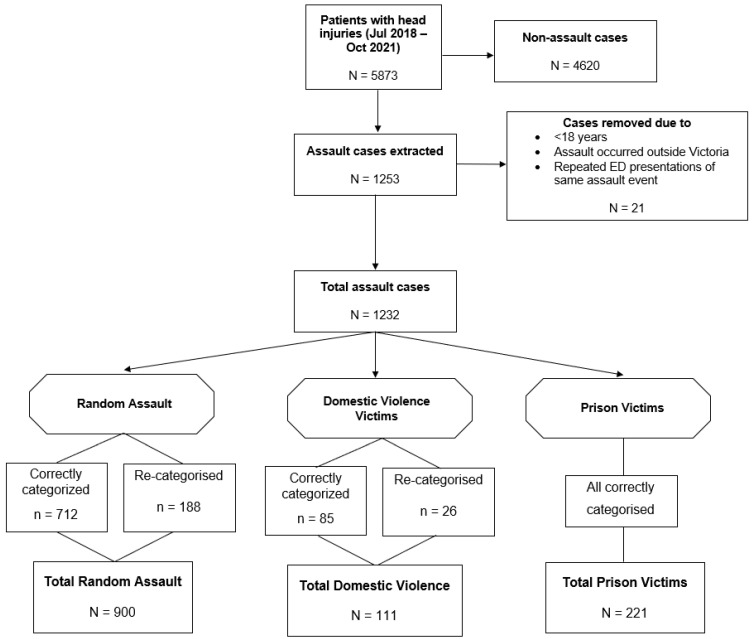
Flowchart of study sample selection along with inclusion and exclusion criteria.

**Figure 3 ijerph-20-00063-f003:**
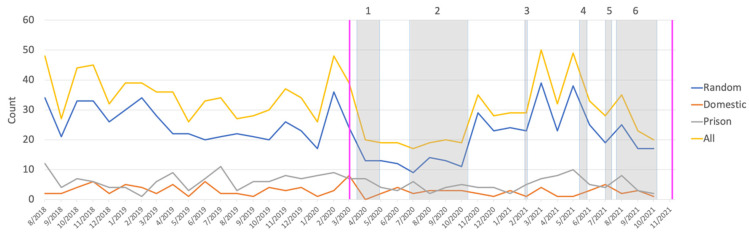
Time series showing the monthly number of assault cases for every assault group and in total. Onset and end of restrictions are indicated by magenta reference lines. Numbered shaded areas represent the Melbourne lockdowns.

**Figure 4 ijerph-20-00063-f004:**
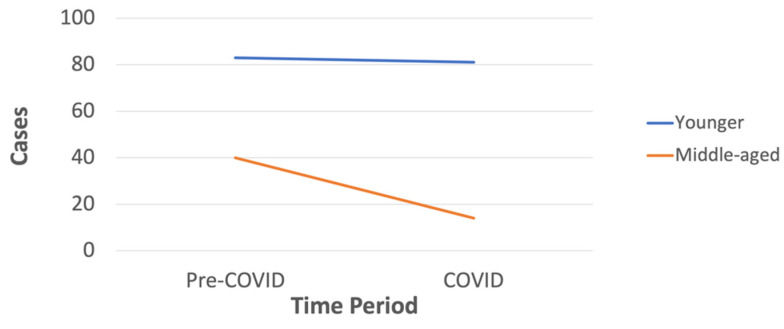
Change in assault-related TBI cases by age group for prison victims between pre-COVID and COVID periods.

**Figure 5 ijerph-20-00063-f005:**
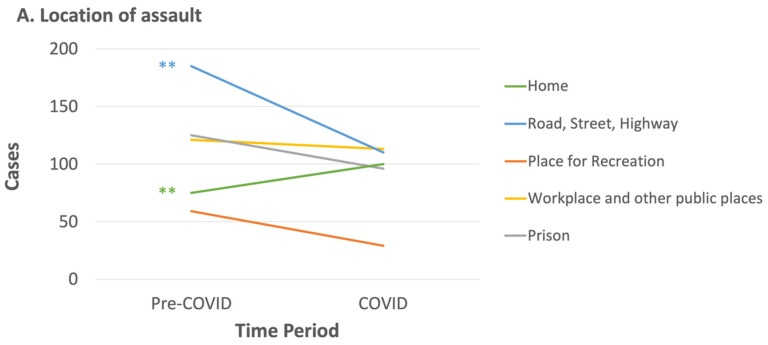
Change in assault-related TBI cases by location of injury between the pre-COVID and COVID periods: (**A**) across all locations of injury (** significant adjusted residuals at *p* < 0.05, FWE corrected); (**B**) across random and domestic assault groups at home in middle-aged (40–59 years old) adults.

**Figure 6 ijerph-20-00063-f006:**
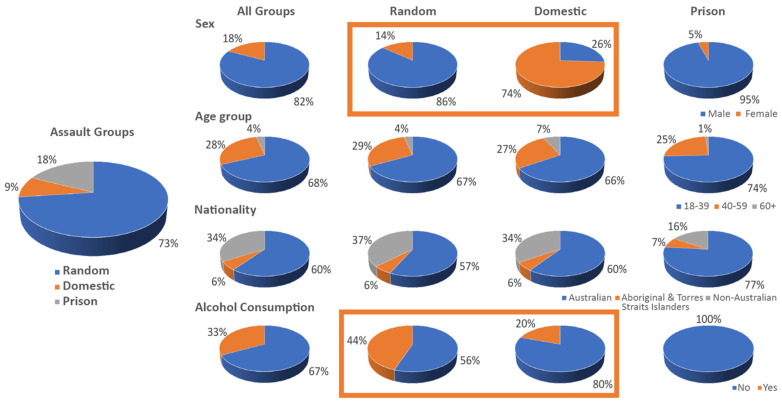
**Who** was a victim of assault-related TBI in Melbourne, Australia, between 30 July 2018 and 30 October 2021? Left, overall proportion per assault group. Right, breakdown of assault-related TBI by sex, age group, nationality, and alcohol consumption (rows) and assault group (columns). Highlighted in orange squares are noticeably contrasting differences in the proportion of cases between RV and DV assault in terms of sex and alcohol consumption. percentages in nationality and alcohol consumption exclude ‘Unknown’ cases (see Appendix A).

**Figure 7 ijerph-20-00063-f007:**
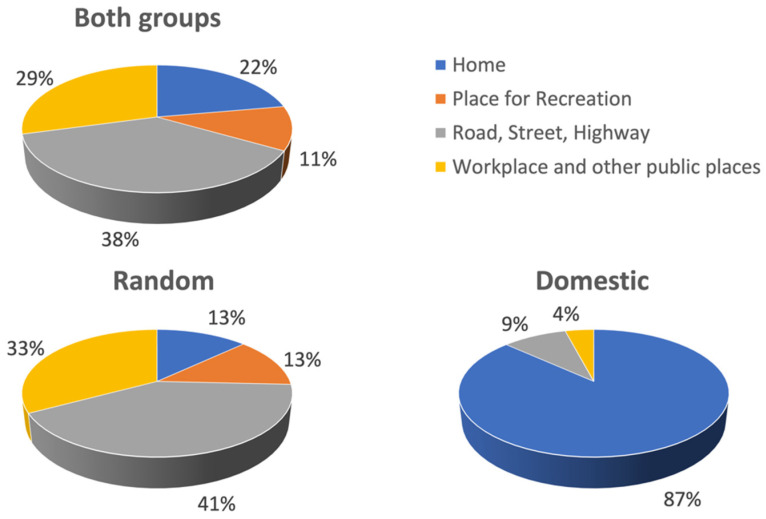
**Where** were victims of assault-related TBI in Melbourne, Australia, attacked between 30 July 2018 and 30 October 2021 (excludes PP victims)?

**Figure 8 ijerph-20-00063-f008:**
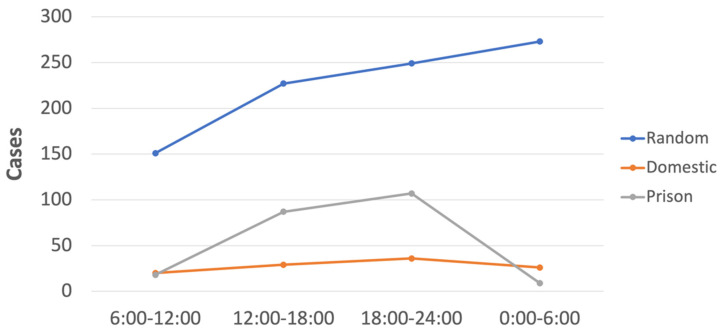
**When** did victims of assault-related TBI in Melbourne, Australia, present at the ED between 30 July 2018 and 30 October 2021? Shown in the plot are number of cases across morning (6:00–12:00), afternoon (12:00–18:00), evening (18:00–24:00), and night (0:00–6:00) periods for the random, domestic, and prison assault subgroups.

**Table 1 ijerph-20-00063-t001:** Dates and duration of COVID-19 restrictions and lockdowns in Melbourne, Australia.

Event		Dates	No. of Days
Restrictions	Start	16 March 2020	-
Lockdown	1	30 March to 12 May 2020	43
Lockdown	2	8 July to 27 October 2020	111
Lockdown	3	12 February to 17, 2021	5
Lockdown	4	27 May to 10 June 2021	14
Lockdown	5	15 July to 27 July 2021	12
Lockdown	6	5 August to 21 October 2021	77
Restrictions	End	19 November 2021	-
**Total–lockdowns**	**6 ^1^**	**-**	**262 ^2^**
**Total–restrictions**	**-**	**-**	**613 ^3^**

^1^ Total No of lockdowns, ^2^ Total No of lockdown days, ^3^ Total No of restrictions days.

**Table 2 ijerph-20-00063-t002:** Demographic characteristics between COVID periods across all assault subgroups.

Demographic Variables	Pre-COVID (n = 689) n (%)			COVID Period (n = 543) n (%)		
	Random	Domestic	Prison	Random	Domestic	Prison
**All**	502 (72.9)	62 (9.0)	125 (18.1)	398 (73.3)	49 (9.0)	96 (17.7)
**Sex**						
Male	432 (86.1)	16 (25.8)	120 (96)	344 (86.4)	13 (26.5)	91 (94.8)
Female	70 (13.9)	46 (74.2)	5 (4)	54 (13.6)	36 (73.5)	5 (5.2)
**Age Group**						
18–39	336 (66.9)	38 (61.3)	83 (66.4)	269 (67.6)	35 (71.4)	81 (84.4)
40–59	143 (28.5)	19 (30.6)	40 (32)	118 (29.6)	11 (22.5)	14 (14.6)
60+ ^a^	23 (4.6)	5 (8.1)	2 (1.6)	11 (2.8)	3 (6.1)	1 (1)
**Nationality**						
Australian	301 (60.0)	38 (61.3)	87 (69.6)	234 (58.8)	32 (65.3)	60 (62.5)
Indigenous ^b^	25 (8.3)	4 (10.5)	6 (6.9)	30 (12.8)	3 (9.4)	8 (13.3)
Non-Australians	194 (38.6)	24 (38.7)	14 (11.2)	154 (38.7)	16 (32.7)	17 (17.7)
Unknown ^c^	7 (1.4)	0 (0)	24 (19.2)	10 (2.5)	1 (2)	19 (19.8)
**Alcohol Consumption** ^d^						
No	147 (55.3)	29 (82.9)	81 (100)	119 (60.1)	22 (75.9)	49 (100)
Yes	95 (35.7)	3 (8.6)	N/A	66 (33.3)	3 (10.3)	N/A
Unknown ^c^	24 (9)	3 (8.6)	0 (0)	13 (6.6)	4 (13.8)	0 (0)

^a^ Removed in chi-square analysis due to low cell count. ^b^ Aboriginal and Torres Strait Islanders. Percentages are based on Australian group rather than assault subgroup totals. ^c^ ‘Unknown’ subgroup removed from chi-square analyses. ^d^ Data restricted to one year of pre-COVID (n = 382) as data collection halted one year into the COVID period (n = 276; total n = 658).

**Table 3 ijerph-20-00063-t003:** Assault-related TBI cases by location of injury during the pre-COVID and COVID periods.

Location of Injury	Pre-COVID	COVID Period
Home	75 (10.9)	100 (18.4)
Road/Street/Highway	185 (26.8)	110 (20.2)
Place for Recreation	59 (8.5)	29 (5.3)
Workplace and other public places	121 (17.5)	113 (20.8)
Prison	125 (18.1)	96 (17.6)
Unknown ^a^	126 (18.2)	96 (17.6)

^a^ ‘Unknown’ subgroup removed from chi-square analyses.

**Table 4 ijerph-20-00063-t004:** Time variables between COVID periods across all assault subgroups.

Time Variables	Pre-COVID n (%)			COVID Period n (%)		
	Random	Domestic	Prison	Random	Domestic	Prison
**ED Presentation Time**						
Morning (6:00–12:00)	83 (16.5)	12 (19.3)	9 (7.2)	68 (17.1)	8 (16.3)	9 (9.4)
Afternoon (12:00–18:00)	124 (24.7)	17 (27.4)	51 (40.8)	103 (25.9)	12 (24.5)	36 (37.5)
Evening (18:00–24:00)	132 (26.3)	19 (30.6)	59 (47.2)	117 (29.4)	17 (34.7)	48 (50)
Night (0:00–6:00)	163 (32.5)	14 (22.6)	6 (4.8)	110 (27.6)	12 (24.5)	3 (3.1)
**TOI Reporting > 24 h**						
No	236 (47)	25 (40.3)	78 (62.4)	197 (49.5)	23 (46.9)	58 (60.4)
Yes	45 (9)	5 (8.1)	10 (8)	35 (8.8)	7 (14.3)	2 (2.1)
Unknown ^a^	221 (44)	32 (51.6)	37 (29.6)	166 (41.7)	19 (38.8)	36 (37.5)

^a^ ‘Unknown’ subgroup removed from chi-square analyses. ED, emergency department; TOI, time of injury.

## Data Availability

The data presented in this study are available on request from the corresponding author.

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
