# Peer review of "Effects of the Response to the COVID-19 Pandemic on Assault-Related Head Injury in Melbourne: A Retrospective Study"

_ijerph, 2022, doi:10.3390/ijerph20010063_

Round 1

Reviewer 1 Report

In this manuscript, the authors examine the effect of the pandemic restrictions on potential changes in assault-related head injury cases. This was accomplished by assessing details about who, where, and when of assault-related head injuries cases that presented at an ED of an inner-city hospital in Melbourne. In particular, the who was determined by sex, age group, nationality, and alcohol consumption; the where by the location at which assault took place: home, road/street, place of recreation, workplace and other public areas, prison, unspecified and the when by the time of day presenting, whether it was within 24 hours of injury and whether it was pre- or post-lockdown. Three subgroups were analysed: random violence (RV); domestic violence (DV); prison population violence (PP) based on the narrative recorded during patient interviews by the triage nurses. Results showed that the lockdown did not significantly affect the proportion of cases by assault subgroup or when during the day they occurred; only exception was significantly lower number of middle-aged PP assault victims during lockdown compared to before. It was also demonstrated that home assaults were significantly increased during lockdown and street assaults significantly decreased during lockdown. The former was due mainly to an increase in the number of random assaults happening to middle-aged adults at home. Based on the relatively small number of changes prior to vs. after the lockdown, the authors also looked more generally at overall trends and found that males were much more likely to be victims of RV assault cases whereas females were much more likely to be victims DV assault cases were female.

This is a well-written paper that uses appropriate methods to gain insight into potential changes in different types of assault-related head injuries resulting from pandemic restrictions. The authors provide appropriate background information to clearly contextualize the study and their findings and the results will add to our knowledge of different types of assault-related head injury and how the public health restrictions may have affected them. I have several comments the authors may wish to address in a revision:

1.     The main unique and somewhat controversial finding is that there was an increase in the number of random assaults resulting in head injuries occurring in the home after the lockdowns were enacted (at least in middle-aged people). This is difficult to understand and the authors suggest that one potential explanation is that these cases were likely mostly DV for which the narrative was insufficient to categorize as such. It may make more sense to make this distinction formal and re-run the analysis with these cases removed or included as a separate group.

2.     More justification for inclusion of the prison population appears warranted. At some level this appears to be a comparison group and I wonder if a bit more background information about why this is a good comparison group would help clarify their inclusion.

3.     Including information on any changes in policy by the ED regarding accepting patients relative to their injury severity given the challenges associated with the pandemic and the health care system’s response to it should be included. If there is information on injury severity included in the files that would be useful to add to the analysis. One can imagine that people may be less likely to show up at the ED during a pandemic because of the fear of becoming infected and the hospital may have put policies in place to protect potential patients and their staff for this reason. The resulting effect may have been that only those with more severe injuries may have made use of the ED.

4.     The introduction does a good job of discussing the heterogeneity of previous work regarding both increases and decreases at least for DV cases. I think it would be beneficial to include some additional narrative in the discussion that reflects back on this heterogeneity specifically as it relates to the different methods used to collect relevant data (i.e., patient records from an ED vs. reports of calls to DV shelters for example).

5.     If there is any information in the triage reports about how the patients were transported to the ED (i.e., via ambulance vs. driven by family/friends) that may provide some insight into both injury severity and barriers facing patients who have been assaulted under the different categories.

6.     If there is sufficient data, it may also be interesting to breakdown the pre/post lockdown periods into different time periods – for example, 10-19.5 months before the lockdown it is likely that most people were not even aware of the coming pandemic, whereas during the few months leading up to the lockdown, it was likely most were starting to feel more and more stress about the situation. Similarly, during the lockdown and the different phases of it, people’s responses were likely impacted by changes in the levels of restrictions in a manner which could have had an influence on the likelihood of violence occurring.

Author Response

Please see the attachment review report 1.docx

Reviewer 2 Report

The authors have presented a well-balanced analysis of assault-related, hospital-treated TBI prior to and during the COVID lockdowns. This is an interesting and worthwhile investigation. The comments I have included below are specifically from a family violence perspective.

Introduction

Throughout the introduction, the authors have highlighted the heterogeneity of prevalence estimates for DV. This is going to be based on a number of different factors, as observed by the authors. Given this, I think it would be helpful to note differences in data sources when referring to increases/decreases. For example, lines 72-75 and lines 106 to 120.

Methods

The authors have made some interesting decisions with the potential to have a particular impact on prevalence estimates for DV. For example, why were only those aged over 18 years included? Children are also at risk of escalating violence during lockdown.

There is also an indication that readmissions for the same assault injury were not included. It is important that the authors allow for subsequent admissions for new injuries, and I would expect that this would be the case, but it would be helpful to have it clarified.

Line 184 - could you please provide a description of the 'human intent descriptor'?

Line 191 - 'unambiguous mention of assault' - I understand that this aligns with ICD coding rules. However, it is not unheard of for a DV victim to present to the ED, describing the injury as unintentional and for hospital staff to note '? assault' in accompanying narratives. Was there any consideration of these cases?

Results

The tables need to be placed closer to where they are referenced in the text.

Figure 5: I am not sure this an effective way of displaying this information. It is difficult to see the comparisons. I also suspect some of the frequency counts are getting small and this display doesn't allow an acknowledgement of that.

Lines 301-303: 'Random violence' occurring at home - it would be helpful to understand these more. Do we understand the nature of the relationship between the offender and the victim?

Table 2: 'Unknown' removed from chi-squared analysis. I would be tempted to keep this category in. It holds meaning. Similar for table 3 - the fact that it is not known indicates the injured person either wasn't capable or didn't feel safe describing when and where. These are significant issues for the DV field.

Discussion

Lines 387-392: The authors might want to consider the literature on child to adult violence. While described as 'random' and being inflicted by a 'stranger', there is the potential for this to be intrafamilial violence and for the victim to be feeling protective (especially where there is the potential for parental stress influencing a young person's use of violence, or the violence being a defensive action to protect another parent).

Author Response

Please see the attachment review report 2.docx
